# A Survey on the Security Challenges of Low-Power Wireless Communication Protocols for Communicating Concrete in Civil Engineerings

**DOI:** 10.3390/s23041849

**Published:** 2023-02-07

**Authors:** Gaël Loubet, Eric Alata, Alexandru Takacs, Daniela Dragomirescu

**Affiliations:** 1LAAS-CNRS, Université de Toulouse, CNRS, INSA, 7, Avenue du Colonel Roche, 31400 Toulouse, France; 2LAAS-CNRS, Université de Toulouse, CNRS, UPS, 7, Avenue du Colonel Roche, 31400 Toulouse, France

**Keywords:** Wireless Sensor Network (WSN), Cyber-Physical System (CPS), cyber-security, Internet of Things (IoT), Structural Health Monitoring (SHM), data integrity, data confidentiality

## Abstract

With the increase in low-power wireless communication solutions, the deployment of Wireless Sensor Networks is becoming usual, especially to implement Cyber-Physical Systems. These latter can be used for Structural Health Monitoring applications in critical environments. To ensure a long-term deployment, battery-free and energy-autonomous wireless sensors are designed and can be powered by ambient energy harvesting or Wireless Power Transfer. Because of the criticality of the applications and the limited resources of the nodes, the security is generally relegated to the background, which leads to vulnerabilities in the entire system. In this paper, a security analysis based on an example: the implementation of a communicating reinforced concrete using a network of battery-free nodes; is presented. First, the employed wireless communication protocols are presented in regard of their native security features, main vulnerabilities, and most usual attacks. Then, the security analysis is carried out for the targeted implementation, especially by defining the main hypothesis of the attack and its consequences. Finally, solutions to secure the data and the network are compared. From a global point-of-view, this security analysis must be initiated from the project definition and must be continued throughout the deployment to allow the use of adapted, updatable and upgradable solutions.

## 1. Introduction

In the past years, climate change has led to a re-consideration of the fabrication and transportation processes, especially with the aims of minimizing carbon footprint and of using renewable energies. These global demands require the introduction of new paradigms, new materials, new energy sources, and new fabrication techniques. This is particularly true in the construction and civil engineering industries, where sustainability, maintainability, and reliability are required for structures and infrastructures [1]. To deal with these new demands, the use of Structural Health Monitoring (SHM) solutions is favored [2]. These consist of an autonomous and “permanent” inspection of the health of the structure to achieve intelligent data-driven diagnostics, and thus to prevent its irreversible failures, avoid its collapse, and allow preventive maintenance to be applied.

Cyber-Physical Systems (CPS), in the framework of the Internet of Things (IoT), are good candidates to implement the Structural Health Monitoring of civil engineering structures [3]. Indeed, these can monitor and/or control the physical world, as well as connect the physical and digital worlds (for instance by updating the digital/virtual models/twins/representations with the data collected by the nodes, but also by commanding the nodes based on the needs of the digital/virtual models/twins/representations). The physical part of the Cyber-Physical Systems can be based on the use of Wireless Sensor Networks (WSN) [4], which are able to wirelessly exchange (with the humans and/or machines) data commonly generated with Non-Destructive Testing (NDT) methods that do not alter the element under test [1,5,6]. With the digitalization and the miniaturization of electronics, this kind of embedded systems is always more effective and pervasive. Nevertheless, the long-term deployment of Wireless Sensor Networks is today mainly restricted by energy autonomy. To lengthen their limited lifespan, ambient energy harvesting and Wireless Power Transmission (WPT) solutions are investigated to power these [7,8]. By considering both Wireless Power Transmission and wireless communication, Wireless Sensor Networks can answer to the Simultaneous Wireless Information and Power Transfer (SWIPT) paradigm [9].

In addition, the cyber-security considerations (especially in terms of data integrity, availability, and confidentiality, but also of alteration or interruption of service) are usually not addressed during the design and implementation phases of the Cyber-Physical Systems and of its Wireless Sensor Network, but only *a posteriori,* or when necessary (e.g., after an attack, or an attempt of attack). However, the hardware and software solutions to protect a Cyber-Physical System have a significant cost, in terms of energy consumption and money, which must be considered at the earliest stages of a project [10].

In this paper, the low-level security analysis of a Wireless Sensor Network conducted during the McBIM project [11,12,13,14] will be presented. This project aims to propose an implementation of the concept of “communicating materials” [15] in the case of reinforced concrete, in part to ensure the Structural Health Monitoring of reinforced concrete structures thanks to Non-Destructive Testing methods based on the use of dedicated Wireless Sensor Networks.

Section 2 will address the designs of the proposed Cyber-Physical System, and of its Wireless Sensor Network based on Communicating Nodes (CN) and Sensing Nodes (SN). The Section 3 will deal with the presentation of the used wireless communication technologies namely LoRa/LoRaWAN and Bluetooth Low Energy (BLE), especially in regard of their native security features, their main vulnerabilities, and their common attacks. Section 4 will present the security analysis carried out during the McBIM project [11], by explicating the potential malicious objectives, the threat model, and the risks. Before concluding, Section 5 will propose the analysis of technical solutions used to prevent the attacks or limit their effects, and thus, to mitigate the risks.

## 2. Architecture of a Cyber-Physical System to Implement a Communicating Concrete

The proposed Cyber-Physical System, presented in Figure 1 and in detail in [11,12,13,14], is composed of a Wireless Sensors Network based on Communicating Nodes (CN) and Sensing Nodes (SN), organized in a two-levels network. Each element made of communicating concrete embeds at least one Communicating Node and several Sensing Nodes, this association forming a subnetwork. Their number is a function of the size of the element and the needs in terms of measurement (e.g., spatial accuracy, etc.).

The Communicating Nodes form an *ad-hoc* mesh network within a structure or a set of adjacent structures. These are intended to aggregate, and then process, store, and share the data transmitted by the Sensing Nodes. The data can be processed, stored, and shared locally in one or more Communicating Nodes of the network, and/or remotely in one or more other networks or even in the digital world (and especially in digital/virtual models/twins/representations), thanks to access to the Internet. Thus, bi-directional medium to long-range wireless communication technologies are required for the communications between the Communicating Nodes. Moreover, at least one Communicating Node per mesh network must be a reliable access point (or a gateway) to the digital world by providing a bi-directional connection to the Internet. Other bi-directional wireless communications technologies can be implemented to interface with other local Wireless Sensor Networks and/or devices. Because these have sufficient energy and processing resources, the Communicating Nodes can employ the usual solutions (e.g., cryptography, etc.) to protect the bidirectional wireless communications, but also the stored data. Thus, the safety aspects concerning the data storage by the Communicating Nodes, but also the wireless communications between the Communicating Nodes and from the Communicating Nodes to the Internet or to extern devices, are not discussed here in order to focus on the wireless communications between the Sensing Nodes (whose the hardware and software architectures are fixed) and the Communicating Nodes.

A star network of Sensing Nodes is available around each Communicating Node, which, thus, becomes a central hub in a subnetwork. The Sensing Nodes are intended to measure relevant parameters of the monitored element and/or its environment (e.g., temperature, relative humidity, mechanical deformation, etc.). The collected and pre-processed data must then be transmitted to the associated Communicating Node(s) with directional medium-range wireless communication technologies (e.g., LoRaWAN or Bluetooth Low Energy) reliable even through the reinforced concretes. In addition to the recovery of the data sent by the Sensing Nodes, the Communicating Nodes have to wirelessly power the Sensing Nodes located in their neighborhood. By tuning their wireless power source (in terms of the waveform, output power, and/or periodicity of activation), the Communicating Nodes can set up the periodicity of functioning of the Sensing Nodes. A radiative electromagnetic Wireless Power Transfer system is used to achieve this Wireless Power Transfer.

The Sensing Nodes are the core elements of this Cyber-Physical System, because gathering the main constraints: inaccessible, energy-autonomous, fully wireless, and long-term usable, resilient, and reliable, during their entire lifetime expressed in decades. Thus, these are designed as simply as possible in order to minimize the risk of failure, and are also battery-free and able to cold-start. In their current implementation, which is fully presented in [12,13,14], the Sensing Node is completely inaccessible and cannot be changed, repaired, or updated. Indeed, there is no physical access (these are encapsulated in the core of the reinforced concrete), nor wireless access (no data downlink is implemented), with an exception for the Wireless Power Transfer that can control the periodicity of activation. It could be noted that in the implementation based on the LoRa technology, a unique antenna is used both for harvesting the electromagnetic power transmitted by the Communicating Node(s) and for sending the collected data to the Communicating Node(s). Due to their limited resources (in terms of processing and energy), their inaccessibility, and their targeted lifespan, the Sensing Nodes are the focus of this low-level security analysis, and in particular their wireless communications with the Communicating Node(s).

Because of the specific implementation and design constraints, this case study of communicating concrete differs from more usual deployments of Wireless Sensors Network, especially because the Sensing Nodes have a very limited (physical and/or wirelessly) and resources (mainly in terms of available energy but also in terms of processing and data storage resources).

Firstly, as there is no physical access to the Sensing Nodes once deployed (as these are encapsulated in reinforced concrete), it is assumed that the attacker cannot physically access it either. Thus, it seems useless to consider the physical attacks for this case study. Nevertheless, this makes impossible to change, repair, update, or upgrade the hardware part of the Sensing Nodes, once deployed and/or after an attack or a compromise.

Then, because of the specific design of the Sensing Nodes: with limited energy resources, limited processing resources, and limited data storage resources, but also without a data downlink; it is assumed that the attacker cannot wirelessly access it to reconfigure or compromise its firmware or exploit its processing resources (e.g., hijacking the network for others activities). Nevertheless, this makes it impossible to employ classic cryptographic solutions (because of the lack of energy and processing resources) but also to update or upgrade the software part of the Sensing Nodes, whose version of the communication protocol or the security algorithms. Thus, the main challenge lies in ensuring the authentication of legitimate frames received by the Communicating Node(s).

Also, as a single and standardized wireless communication protocol is employed, both the attacker and the designer are aware of the attacks detailed in the scientific literature, and the designer must use this knowledge to deploy specific and tailored countermeasures to protect the Sensing Nodes and/or to mitigate the risks.

As the propagation medium of the electromagnetic waves between the Sensing Nodes and the Communicating Nodes (namely the reinforced concrete) is very harsh and greatly attenuates the radiofrequency signals, the attacker can easily overwrite the legitimate wireless communications to craft malicious messages. Nevertheless, the Communicating Nodes can be allowed to use this knowledge to identify malicious frames based on signals with abnormally high-power levels.

Finally, as the Sensing Nodes are remotely and wireless powered by a radiative electromagnetic Wireless Power Transfer solution, a frame reception must be preceded by a power transfer managed by the meshed network of Communicating Nodes. Thus, any frame reception that does not complain about this behavior can be considered suspicious by the Communicating Node(s).

## 3. Low-Level Security Aspects of the LoRaWAN and Bluetooth Low Energy Wireless Communications Protocols

For the next, and because already implemented in the proposed solutions [12,13,14], both the LoRaWAN [16,17] and Bluetooth Low Energy [18,19] wireless communication protocols will be analyzed regarding their low-level security aspects. The implemented security features, their most usual vulnerabilities, and their common attacks will be introduced for each one, even if these are not applicable to the current implementations. Whatever the targeted application and the environment of deployment, all these elements concerning the studied wireless communication protocols must be known and considered. Given the use of standard protocols, it is necessary to continuously conduct the monitoring of technological development on the security aspects of the employed protocols, as these directly impact the object and its use. This is the case here, where all these aspects concerning the LoRaWAN and Bluetooth Low Energy technologies must be considered for Sensing Nodes (whose hardware and software architectures are fixed) communicating from the core of reinforced concretes.

### 3.1. LoRaWAN

Regarding the LoRaWAN wireless communication protocol, the 1.0.3 version of the specification, the use of Class A devices, and the absence of acknowledgment (and more generally of data downlink) will be considered, as currently implemented in the Sensing Nodes [16,17]. Its low-level security aspects are recent research topics [16,17,20,21,22,23,24,25,26].

#### 3.1.1. Native Security Features

A LoRaWAN device has a unique 64 bits identifier (*DevEUI*) and a unique 32 bits address (*DevAddr*), and must be authenticated in order to transmit data to a network. This authentication can be achieved by over-the-air activation (OTAA) (not implemented in the McBIM project) or by activation by personalization (ABP) (implemented in the McBIM project). In both cases, the device obtains two unique AES-128 symmetric session keys named *AppSKey* and *NwkSKey* assigned before data communication for a unique communication session. *NwkSKey* is shared with the network server and is used to calculate and verify the MIC (Message Integrity Code) of all data frames to ensure data integrity; and to encrypt and decrypt the payload field of a MAC (Medium Access Control) data frames. Whilst, *AppSKey* is shared with the application server and is used to encrypt, by an XOR operation, and decrypt the payload field of application-specific data frames. The over-the-air activation procedure ensures unicity for keys by generating these from a unique key named *AppKey*, this at each reset or re-join request; whilst this is the responsibility of the developer in the activation by personalization procedure to ensure the unicity for the static keys assigned and stored directly in the device. This unicity allows to reduce the probability of compromising the whole network while a node is compromising. In order to prevent replay attacks and packet losses, two frame counters can be used to keep uplink and downlink messages synchronized. If the difference between these is greater than a limitation value, the frames are dropped. In the current implementation, the frame counter is disabled for development purposes, because the application deployed on the application server must be updated each time a Sensing Node is programmed with a new firmware. An acknowledgement frame can be sent in response to an accepted uplink frame. If not received, the uplink frame can be retransmitted. After several attempts, the frame can be considered lost or rejected. In the proposed implementation, the acknowledgement is disabled in order to limit energy consumption by not considering the data downlink, whatever its form. Thus, the LoRaWAN specification provides an authentication procedure to join a network; the encryption of the payload based on an Advanced Encryption Standard (AES) algorithm and the use of keys: the *NwkSKey* or the *AppSKey*; and an integrity check of each data frame sent. More, some additional procedures are available to ensure some security functions.

#### 3.1.2. Usual Vulnerabilities

Physical access to devices

By having physical access to devices, it becomes possible to extract the device and network security keys (e.g., through reverse engineering by deriving the key from public information, etc.), especially *AppSKey* and *NwkSKey* which are necessary to decrypt the communications; and thus, to compromise both the device and the network. The consequences are that: the communications could be decrypted; an attacker could create a mock device with the same credentials to impersonate a legitimate device; the data payload can be manipulated; etc. It is also possible to use hardware, especially a radio module, near the targeted device to intercept its communications. To prevent compromises, the critical data should not be shared.

2.Lack of association between frames

One of the most important vulnerability is the lack of association between data frames and their acknowledgements, especially during the over-the-air activation procedure, which promotes replay attacks and acknowledgement spoofing. Two solutions have been implemented: the frame counter is included in the calculation of the message integrity code; and an acknowledgement flag is added.

3.Re-use of nonce values

Nonce values are values pseudo-randomly generated and used only once to derive the security keys during the over-the-air activation procedure. Because not tracked in some versions, there is a risk of generating a value already used, making the network vulnerable to replay attack or eavesdropping. A solution has been implemented: the nonce values are turned into counters; and the last used values are stored and tracked.

4.Frame counter management

When a device is rebooted or when its frame counters overflow, these latter are set to 0. By being able to reset a device, the frames obtained before by sniffing the communications could be replayed back during a replay attack. For the activation by personalization procedure, a solution could be to store the counter values in the server during the reboot of a device, and rejecting all the messages while the new counter does not reach the stored value. This would decrease the availability of the device. For the over-the-air activation procedure, a solution has been implemented: new security keys are generated at each reconnection.

5.Lack of end-to-end integrity protection

The integrity of the application data is not protected during its transmission between the network and application servers. The specifications acknowledge this vulnerability but are left to the developer of the application to implement its own security features.

6.Packet and payload vulnerabilities

The frames are not time-stamped to validate the time of the transmission, which makes it vulnerable to replay attacks. More, its payload length is the same before and after the encryption. Therefore, an attacker could overflow counters to restore the key stream from the encrypted messages.

7.Credentials Misconfiguration

Security of exchanges relies on cryptographic keys embedded in the devices. These cryptographic keys are used to provide authentication of the device and confidentiality of exchanged data. On the other hand, some devices need bi-directional communications which imply the possibility for a remote system to connect to the device using a password. A common mistake, during the deployment of devices, is to reuse the same keys and passwords for all devices. As well, we can consider that these default values are well known to the intruder. For instance, this weakness leads to the mirai worm [27].

#### 3.1.3. Common Attacks

Radio jamming

The radio jamming consists of a malicious entity in transmitting a powerful radio signal near devices and/or gateways, to disrupt the radio transmissions. Because of Chirp Spread Spectrum (CSS) modulation coexistence issues, malicious LoRa transmissions on the same frequency and with the same spreading factor used by the legal LoRa transmissions are sufficient to interfere with these. Almost all the transmissions can be affected and wiped out at the frequency used. This attack can be detected by observing a sudden drop out from the network. Once detected, it is recommended to change the frequency band.

2.Replay attack

During a replay attack, the attacker captures a valid data transmission to repeat or delay it to fool the network (both device and gateway can be targeted). The attack requires knowledge of the frequencies and channels used during the communications. These can be prevented with the use of the tracking frame counters, join procedure via over-the-air activation, or physical protection; and could lead to Denial of Service (DoS) which intends to disrupt services.

3.Acknowledgement spoofing

This attack results from the lack of association between a frame and its acknowledgment. The attacker prevents the reception (e.g., via jamming) and captures the downlink acknowledgement in order to acknowledge another uplink frame from the same device. The purpose of this attack is mainly: to take control of the gateways; to damage the network; or to provoke Denial of Service. This is also possible on uplink frames if the attacker can prevent their reception by gateways.

4.Bit flipping

The lack of end-to-end integrity protection of application data enables bit flipping. If the transport layer security between the network and application servers does not exist or is compromised, and if the attacker is able to act on this channel, then the application data can be altered and the confidentiality of the application compromised.

5.Eavesdropping

Eavesdropping can be passive (e.g., sniffing) or active (e.g., relay attack, man-in-the middle). During the sniffing attack, the most common passive eavesdropping method, the attacker captures the frames transmitted over a network between the devices and the gateways. From the gathered information, the attacker can launch further to compromise the operation of the network at several levels.

6.Relay attack

Relay attack occurs when a malicious entity creates a relay between the devices and the network server, and initiates a communication to relay the frames to another malicious entity.

### 3.2. Bluetooth Low Energy

Regarding the Bluetooth Low Energy wireless communication protocol, the 5th version of the specification and the use of the topology based on broadcasters and observers will be considered, as currently implemented in the Sensing Nodes [18,19]. Its low-level security aspects are recent research topics [18,28,29,30,31,32,33,34,35,36,37,38].

#### 3.2.1. Native Security Features

Several security mechanisms are already implemented by default in Bluetooth Low Energy technology, such as the frequency hopping which avoids interferences with other devices using the same frequency band. More, the implementation of some security processes is recommended in [18,28]. First, two security modes with several levels of security are defined to encrypt and sign data. The mode 1 is dedicated to data encryption. Its level 1 provides no security; its level 2 the unauthenticated pairing with encryption; its level 3 the authenticated pairing with encryption; and its level 4 the authenticated secure connection pairing with encryption. Mode 2 is dedicated to connection-based data signing. Its level 1 and level 2 provide, respectively, the unauthenticated and the authenticated pairing with data signing. Second, a security manager is used for the pairing process during which devices exchange information to establish secure connection (Mode 1, level 4) which avoids temporary key brute-force attacks. The pairing process has three main phases: the exchange (with no encryption) of pairing features (based on the abilities) between the devices in order to select the most suitable method to generate short-term (or temporary) key (four methods are available, namely “just works”, “passkey”, “out-of-bands” and “numeric comparison”); the generation and the exchange of the short-term key used to encrypt the frames dedicated to the pairing and the authentication, and which protect against man-in-the-middle attacks; finally, the generation and the exchange of the long term key used to encrypt all the next communications. An optional phase consists of the exchange of transport key parameters which can be used to store the security keys and the information exchanged during the pairing process, which will allow later re-connections without needing to repeat the entire process. Then, the Bluetooth Low Energy communications are encrypted using an AES-128 cipher block chaining-message authentication code algorithm based on 128 bits key length generated with the elliptic curve Diffie-Hellman method. The communications using encryption and authentication use a Message Integrity Code appended to the payload, and a Cyclic Redundancy Check (CRC) mechanism to protect it all. The communications using authentication but not encryption use a 12-byte signature computed with a 128-bit AES algorithm placed after the data payload, as well as an input counter to prevent replay attacks. Moreover, a privacy feature is provided to limit the tracking of the identity of a device: its address is private and changes frequently, via the encryption of its public address. Finally, trust modes are defined to characterize the communications. Communication with a device “trusted” allows a fixed connection and unrestricted access to all its services, while communication with an “untrusted” device restricts its access to a set of services.

#### 3.2.2. Usual Vulnerabilities

Pairing process

Although the short-term key is not transmitted through the packet, its 16 bytes input value is predictable. For the “just works” method, its value is predefined to ‘0 × 00’ and this method is vulnerable to man-in-the-middle attacks because the authenticity of the connection cannot be verified. For the “passkey” method the generation parameters are transmitted through packets. Thus, an attacker could calculate its value and decrypt data.

2.Discoverability

Bluetooth Low Energy has a discoverability mode used before the pairing process. A discoverable device is vulnerable because it allows all the devices located in its neighborhood to access information, such as its name, its class, and its services. Turning off the discoverability mode prevents devices from scanning attacks.

#### 3.2.3. Common Attacks

Bluetooth Low Energy technology, and more generally Bluetooth technology, is vulnerable to many attacks, whose: the PIN (Personal Identification Number) theft (by cracking or off-line recovery, etc.); eavesdropping (sniffing, man-in-the-middle, relay, etc.); cloning (Medium Access Control address spoofing, forced re-pairing, brute-force, chopping, etc.); the treacherous (backdoor, bumping, etc.); the Denial of Service (radio jamming, Medium Access Control address duplication, synchronous connection-oriented, enhanced synchronous connection-oriented, battery exhaustion, big negative-acknowledgement, guaranteed service, smacking, etc.); the surveillance (printing, stumbling, tracking, etc.); and the miscellaneous others (snarfing, bugging, jacking, free calling, whisperer, etc.); etc. Nevertheless, eavesdropping and Denial of Service attacks are more usual.

Eavesdropping: sniffing

Sniffing is the most common passive approach for eavesdropping. This attack can take place during different stages of Bluetooth Low Energy communication, such as a new connection, an active connection, or a negotiation phase. However, in the case of Bluetooth Low Energy, sniffing is complex and expensive as 40 channels are used with a fast Frequency Hopping Spread Spectrum (FHSS) technique. During the establishment of a new connection, the connection request packet can be captured. This one contains several parameters to set the frequency hopping algorithm and the Cyclic Redundancy Check calculation. Knowing these parameters, the attacker can use these to set up its algorithm to listen from at least one of the three advertising channels, if it is too expensive to sniff all of these at once. During an active connection, the attacker can deduce the connection parameters through an exhaustive approach that assumes that all channels are systematically used, and which is not effective on short communications because a lot of time is required. During the negotiation phase, the attacker can obtain the encryption keys to decrypt the next communications.

2.Eavesdropping: man-in-the-middle

Man-in-the-middle attacks occur when an attacker intercepts the communications between two devices and modifies them. Some attacks consist in cloning the GATT (Generic ATTribute) server to simulate an identical device to which the master device will be connected. It allows the fake device to connect to the legitimate device to capture the traffic, impersonate a device, inject data, modify or redirect packets, provoke Denial of Service, etc. These attacks are easy to implement as only requiring a communication between two devices and as the attacker can negotiate the encryption parameters.

3.Radio jamming

Using a strong radio signal near a Bluetooth Low Energy device can cause interferences and jam communications. The attacker can jam the connected communications and the advertising transmissions by saturating the radio spectrum, until interrupting connected communications or hijacking connected communications by forcing the master to disconnect. Preventing radio jamming is difficult as it requires physical protection from interference.

4.Other attacks

Bluetooth Low Energy is also vulnerable to replay attacks, relay attacks, and spoofing attacks.

5.Audit tools

Several audit tools, such as [39], exist to test the resistance to attacks of the devices under test.

## 4. Security Analysis and Threat Models for Reinforced Concrete Structural Health Monitoring Applications

Both the LoRaWAN and Bluetooth Low Energy wireless communication protocols will be studied regarding their current implementation in the framework of the McBIM project [12,13,14]. The wireless communications between inaccessible Sensing Nodes encapsulated in the reinforced concrete and accessible Communicating Nodes located on the surface of the reinforced concrete will be mainly considered. These are currently only unidirectional from the Sensing Nodes to the Communicating Nodes and carry non-critical measurement data. More, the Sensing Nodes are not able to receive downlink frames but can be controlled by the Communicating Nodes through the Wireless Power Transfer system. Then, the bidirectional communications within the mesh network of Communicating Nodes and with the Internet will be only skimmed through. In any case, all wireless communications: from the Sensing Nodes to the Communicating Nodes; between the Communicating Nodes; and between a Communicating Node and the Internet; raise low-level security issues, whose importance depends on the data transmitted: their type, their criticality, their reliability, etc. Thus, a security analysis could be achieved for each of these wireless interfaces, but also on the hardware side of the Sensing Nodes and the Communicating Nodes, for instance by applying a Failure Mode, Effects, and Criticality Analysis (FMECA) [40].

### 4.1. Malicious Objectives

Three main types of malicious objectives have been identified for the targeted application: the invasion of privacy; the alteration of service; and the interruption of service. Because of low computing resources, the hijacking of the network for others activities (e.g., mining cryptocurrencies, launching a Denial of Service (DoS) attack, etc.) seems improbable.

#### 4.1.1. Invasion of Privacy

Invasion of privacy consists in gathering information on the activities in the instrumented infrastructure, for instance through sniffing or other eavesdropping techniques. This could be realized by the infrastructure owner to acknowledge, for instance, the movements or activities of the users (such as employees, etc.) in the infrastructure. An outsider of the structure could also gather information on activities in order to identify the best moment to trespass in the infrastructure (such as for robbing or degrading, etc.) or to collect classified information (such as the use of the infrastructure, the available equipment, etc.).

#### 4.1.2. Alteration of Service

The services delivered by the communicating reinforced concrete can be altered by the falsification of the measurement, for instance through man-in-the-middle attacks, relay attacks, or replay attacks; or by modifying the transmitted frames. As an example, an attacker could emulate a failure (such as a significant crack, a fire, etc.) to make people believe in the possible collapse of the infrastructure or at least its unsafety.

#### 4.1.3. Interruption of Service

The services delivered by the communicating reinforced concrete can be interrupted by stopping the communications, for instance through Denial of Service attacks, radio jamming attacks, or battery exhaustion attacks (such as by avoiding the Wireless Power Transfer from the Communicating Nodes to the Sensing Nodes, etc.).

### 4.2. Threat Models

The proposed threat model is based on two-range attacks: the short-range and the long-range.

#### 4.2.1. Short-Range Attack

The short-range attacks provide physical access to the attacker which can be either inside the infrastructure or outside it but near enough to place malicious objects (such as malicious Sensing Nodes, malicious Communicating Nodes, etc.). Nevertheless, the Sensing Nodes are considered physically and wirelessly inaccessible.

#### 4.2.2. Long-Range Attack

The maximum range of the attacks depends on the wireless communication technology, the transmission power, and the type of communication. In this case, the attacker is able to communicate with legitimate nodes or to emit an enough powerful radio signal to jam the wireless communications, but also to control the periodicity of activation of the Sensing Nodes by employing its own radiative electromagnetic power source(s).

### 4.3. Risks

The risk scales, both for the probability and the impact of an attack, are based on personal estimations related to the state of the art available in the scientific literature and have been the subject of a consensus among a dozen of experts from the security and different technical fields, working on the McBIM project. The impact of an attack depends on the potential harm this can inflict both to the material and the humans, due to the failure of its detection.

#### 4.3.1. Invasion of Privacy

The invasion of privacy implies several risks such as surveillance, the insertion of a malicious node into the network, the insertion of fake data, and the compromise of node(s). Their analysis is proposed in Table 1.

#### 4.3.2. Alteration of Service

The alteration of service implies several risks such as the deduction of the infrastructure activities or the alteration of data; and is time-consuming and expensive to detect and correct. Their analysis is proposed in Table 2.

#### 4.3.3. Interruption of Service

The interruption of service implies several risks such as radio jamming, the battery exhaustion, the creation of relays, the creation of cycles, the damage of the rectenna, data recovery from nodes, and the alteration of the full infrastructure; and is time-consuming and expensive during the time of unavailability. Their analysis is proposed in Table 3.

## 5. Additional Technical Solutions

In addition to native security features, four main technical solutions can be employed separately or conjointly to prevent the attacks: cryptography [41,42]; Secure Element (SE) [43]; Intrusion Detection System (IDS) [44,45]; and multilayer signature [46].

### 5.1. Cryptography

The wireless communications can be secured by employing the cryptography features offered both by the LoRaWAN and the Bluetooth Low Energy protocols, especially through the encryption of the data, respectively thanks to an AES-128 counter algorithm and an AES-128 cipher block chaining-message authentication code algorithm. Nevertheless, additional levels (s) of cryptography can be employed. The use of cryptography is a flexible solution easy to implement, but it is computationally expensive and which requires secrets to be stored in a safe and non-volatile manner. Moreover, it is power-consuming and generating of latency, thus poorly suited to battery-free low-energy devices.

For instance, and from preliminary experimentations, the power consumption of an NXP QN9080 all-in-one module (MicroController Unit (MCU) and Bluetooth Low Energy transceiver) [47] powered at 1.8 V and designed to achieve (in a suboptimal broadcaster configuration: a start-up, a temperature measurement with the internal sensor, and the transmission of three 21-bytes long advertising frames in 3 different channels (36, 37 and 38) at +0 dBm), can be drawn by almost 90% only by disabling the Security Libraries (SecLib) and mainly the Random Number Generation (RNG) module, even if these are only initialized and never used. Thus, the duration of the process can be reduced from 2.7 s to 355 ms, and the energy needed from 7.9 mJ to 731 µJ.

### 5.2. Secure Elements

The use of Secure Elements can be an alternative way to secure the Wireless Sensor Network [39]. This one is tamper-resistant hardware embedded chip used to secure the storage of confidential and cryptographic data, to host securely applications, and to implement end-to-end security. Resistive Random-Access Memory (RAM) Physical Unclonable Functions (PUF) can be implemented to manage the authentication, the key generation and the storage. The Secure Elements are relatively cheap and consume less energy than using software cryptography. However, its driver (used to manage the communication between the micro-controller unit and the secure element) must be deployed within the microcontroller. Finally, the wire connection to the microcontroller unit must be protected.

### 5.3. Intrusion Detection System

An Intrusion Detection System can be another alternative way to secure the Wireless Sensor Network. This one is based on two detection methods: the signature-based and the anomaly-based methods. The first is not yet adapted for the low-power Wireless Sensor Networks, as we do not yet have enough knowledge of malicious behavior to propose a database of signatures of malicious activities. The second method uses learning systems to model the legitimate behaviors and detect the suspicious behaviors, by comparing observation with the reference model. In the McBIM project, two learning phases can be imagined: one during the manufacture of an element made of communicating reinforced concrete, during which the Communicating Node(s) detect the legitimate Sensing Nodes in its neighborhood; and the other during the construction of a complete structure made of several elements, during which each Communicating Node detects the legitimate Communicating Nodes in its neighborhood. The intrusion detection systems provide visibility on the network and add a layer of defense, but require maintenance and can be sensitive to false positives and negatives.

### 5.4. Multilayer Signature

The multilayer signature (sometimes called *footprint* or *fingerprint*) tends to use a singularity of each communicating object to characterize it and certify the authenticity of its communications in the framework of an Attack Detection System (ADS). This signature can be defined from the hardware (e.g., by the use of a metasurfaces antenna [48]) or the embedded software.

### 5.5. Implementable Features

Table 4 and Table 5 gather some optional security features respectively provided by the LoRaWAN and the Bluetooth Low Energy protocols, with the attacks this prevents and its drawbacks. Just because some attacks are avoided does not mean that there are no more risks.

### 5.6. Security Recommendations and Perspectives in the Case of Communicating Concrete

As a result of the security analysis, it appears that the Sensing Nodes can be considered as always intact over time (unalterable because both their hardware and software are inaccessible and fixed). However, an attacker could still be able to disrupt wireless communications despite the implementations of all the countermeasures presented in this section. Nevertheless, it is possible to mitigate the risks and their consequences. Indeed, the Communicating Nodes have knowledge about the network topology, the context of deployment, the targeted application, but also the behaviors of each Sensing Node. These can also be enriched with a behavioral Intrusion Detection System allowing the detection and identify attacks. Thus, the implementation of a least a behavioral Intrusion Detection System seems essential and this solution can be easily updated and does not impact the architecture and implementation of the Sensing Nodes, but also the architecture and hardware implementation of the Wireless Sensor Network. Moreover, this behavioral Intrusion Detection System can also be deployed in the digital world (and in the digital/virtual models/twins/representations).

## 6. Conclusions

This paper presents the security analysis carried out in the framework of the McBIM project which aims at implementing a communicating reinforced concrete based on a Wireless Sensor Network using wirelessly powered battery-free nodes with low resources (energy, processing, storage). Firstly, the implemented Cyber-Physical System is presented, as well as the employed wireless communication protocols, namely LoRaWAN and Bluetooth Low Energy, in regards to their native security features, their main vulnerabilities, and their most usual attacks. Then, a focus on the issues specific to the proposed implementation is achieved, especially by defining the current implementation, the main hypothesis of attack, and their consequences (from the invasion of privacy to alteration or even interruption of service). The unidirectional wireless communications from the Sensing Nodes (wirelessly powered, battery-free, and low resources) to the Communicating Node(s) are mainly considered, even if other wireless communications are implemented in the proposed Cyber-Physical System (especially in a mesh network of Communicating Nodes, with the Internet, or with local wireless devices such as smartphones), and attacks based on direct access to the Sensing Nodes are not considered as these are assumed to be physically and wirelessly inaccessible (encapsulated in the reinforced concrete and without data downlink). The solutions to secure both the data and the network are studied, in particular, those provided by the considered standards but also those that are in the state-of-the-art, and considered in regards to the available resources (energy, processing, storage). Finally, even if several solutions are implementable *a posteriori*, these must be studied and used from the beginning of the implementation and thought with a global point-of-view. In the presented case study, the use of a behavioral Intrusion Detection System (deployed both in the Communicating Nodes and in the digital/virtual models/twins/representations) seems to be a relevant solution to mitigate the risks.

## Figures and Tables

**Figure 1 sensors-23-01849-f001:**
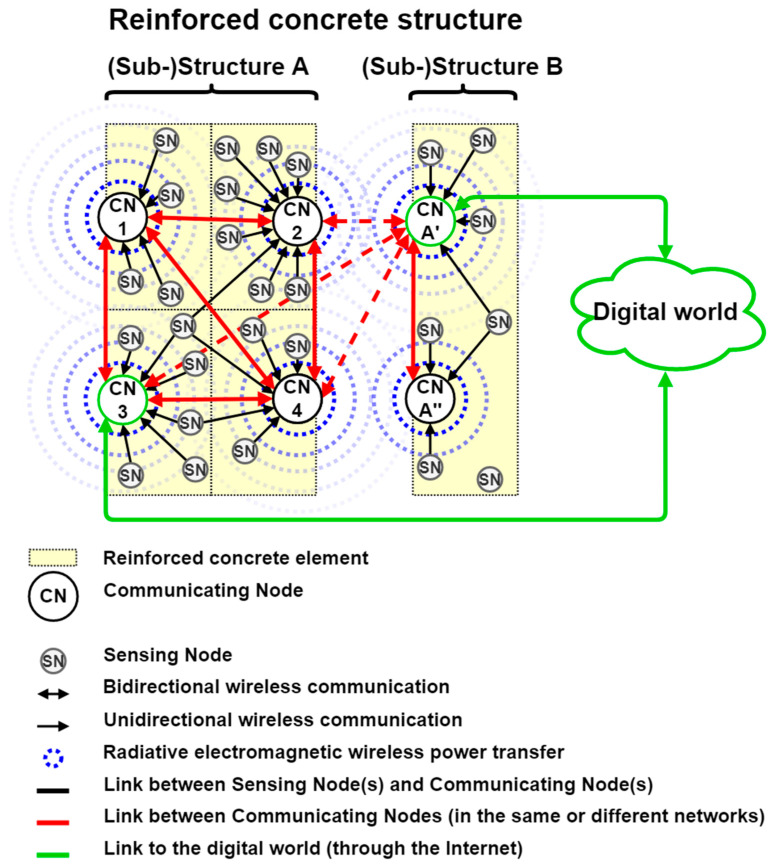
Block diagram of the architecture of the Cyber-Physical System dedicated to the implementation of communicating concrete.

**Table 1 sensors-23-01849-t001:** Analysis of the risks implied by an invasion of privacy.

Risk	Probability	Impact
**Surveillance**	**Likely**	**Insignificant to critical,** depends on the activities
**Insertion of a malicious node into the network**	Sensing Node embedded in the reinforced concrete: **Unlikely**	Sensing Node embedded in the reinforced concrete: **Minor**
	Sensing Node non-embedded in the reinforced concrete: **Likely**	Sensing Node non-embedded in the reinforced concrete: **Minor**
	Communicating Node: **Even**	Communicating Node: **Major to critical** (especially if a gateway to the Internet is targeted)
**Insertion of fake data**	**Likely**	**Moderate to critical**
**Compromise of node(s)**	Sensing Node: **Unlikely**	Sensing Node: **Minor**
	Communicating Node: **Likely**	Communicating Node: **Major to critical** (especially if a gateway to the Internet is targeted)

**Table 2 sensors-23-01849-t002:** Analysis of the risks implied by an alteration of service.

Risk	Probability	Impact
**Deduction of the** **infrastructure activities**	**Likely** depends on the implemented security mechanisms	**Minor to critical** depends on the activities (e.g., critical in a nuclear plant, etc.)
	**Even** depends on the implemented security mechanisms	**Moderate to critical** (e.g., emulation of a failure, a collapse, a fire, etc.)

**Table 3 sensors-23-01849-t003:** Analysis of the risks implied by an interruption of service.

Risk	Probability	Impact
**Radio jamming**	Between the Sensing Nodes and the Communicating Nodes: **Likely**	Between the Sensing Nodes and the Communicating Nodes: **Insignificant to critical** depends on the number of affected nodes
	Between the Communicating Nodes: **Likely**	Between the Communicating Nodes: **Moderate to major**
	Between a Communicating Node and the Internet: **Likely**	Between a Communicating Node and the Internet: **Critical**
**Battery exhaustion**	Alteration of the Wireless Power Transfer: **Improbable**	Wireless Power Transfer: **Critical**
	Destruction of the Sensing Nodes components: **Unlikely** (e.g., mechanical break, etc.)	Sensing Nodes: **Insignificant to critical** depends on the number of affected nodes
	Destruction of the Communicating Nodes: **Even**	Communicating Nodes: **Major to critical** depends on the number of affected nodes
**Creation of relays**	Sensing Node embedded in the reinforced concrete: **Unlikely**	Sensing Node embedded in the reinforced concrete: **Minor**
	Sensing Node non-embedded in the reinforced concrete: **Likely**	Sensing Node non-embedded in the reinforced concrete: **Major**
	Communicating Node: **Likely**	Communicating Node: **Major to critical** (e.g., a malicious device takes the place of a failed node)
**Creations of cycles**	**Likely**	**Minor to critical** depends on the type of activities and of data
**Damage to the rectenna**	**Unlikely** (e.g., very energetic electromagnetic wave, etc.)	**Critical**
**Data recovery from nodes**	**Even**	**Minor to critical** depends on the type of data
**Alteration of the full infrastructure**	**Unlikely**	**Critical**

**Table 4 sensors-23-01849-t004:** LoRaWAN security issues and protection mechanisms.

Security Mechanisms	Attacks Prevented	Consequences of a Successful Attack	Drawbacks
**Over-the-air** **activation** **procedure**	Replay attack	Connection of a malicious device to the network server Injection of (fake) data Etc.	Risk of replay attacks reduced but still possible if the reset and overflow of the frame counter are not well considered Increases latency Increases power consumption Requires data downlink
**Frame counter**	Replay attack	(Re)Use of a valid message to connect a malicious device to the network server (Re)Injection of (fake) data Etc.	Could decrease the availability of a device Reset and overflow must be well considered
**Messageacknowledgement**	Replay attack Acknowledgement spoofing	(Re)Use of a valid message to connect a malicious device to the network server (Re)Injection of (fake) data Etc.	Increases latency Increases power consumption Requires data downlink

**Table 5 sensors-23-01849-t005:** Bluetooth Low Energy security issues and protection mechanisms.

Security Mechanisms		Attacks Prevented	Consequences of a Successful Attack	Drawbacks
**Security Mode 1:** **Encryption**	Level 1	None	Decryption of data Traffic observation Traffic injection Denial of service	N/A
	Level 2	Limited eaves-dropping protection		Requires encrypted link Requires data downlink
	Level 3	Eavesdropping Replay attack		Requires encrypted link Requires data downlink
	Level 4	Eavesdropping Replay attack Man-in-the-middle		Requires encrypted link Requires secure communication Requires data downlink
**Security Mode 2:** **Data signing**	Level 1	None	Traffic observation Traffic injection	Cannot be combined with security Mode 1 Connection-based data signing Requires signing Requires data downlink
	Level 2	Eavesdropping Replay attack		Cannot be combined with security Mode 1 Connection-based data signing Requires signing Requires data downlink
**Pairing process:** **Temporary key generation**	*Just work*	Passive attacks	Impersonate devices Decryption of data Traffic observation Traffic injection Denial of services	Requires data downlink
	*Passkey*	Passive attacks Man-in-the-middle		Input or output ability Requires data downlink
	*Out-of-band*	Passive attacks Man-in-the-middle		Requires another interface Requires data downlink
	*Numeric comparison*	Passive attacks Man-in-the-middle		Requires binary input Requires data downlink
**Discoverable mode disabled**		Prevents from accessing information such as names, class, services, etc.	Theft of sensitive data	No data transmission allowed
**Trust mode**		Limits automatic access to all services	Only a trusted device just compromised enables access to all the services of an attacker	Removes pairing information
**Privacy feature**		Identity tracking	Theft of sensitive data	Available only with connected mode Only a trusted device can be connected

## Data Availability

Not applicable.

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
