# Peer review of "A Survey on the Security Challenges of Low-Power Wireless Communication Protocols for Communicating Concrete in Civil Engineerings"

_sensors, 2023, doi:10.3390/s23041849_

Round 1

Reviewer 1 Report

Security of Wireless Protocols in Critical Environments. Case Study for Communicating Concrete in Civil Engineering.

Consider refining the title to:

Security of Wireless Protocols in Critical Environments - a case study for Communicating Concrete in Civil Engineering.

The title as it stands is a somewhat false premise. The paper surveys as a survey/overview around security challenges relating to (low power) wireless protocols. While this is mentioned in the context of embedded sensors/nodes within concrete structures, there is not an emphasis nor case study of this. This could be addressed by considering a further change in the title relating to claims of communicating concerted

References 1,5,13 deal with aspects of systems in concrete. The abstract claims (lines 21) to be investing in concrete specific aspects.  lines 68/69 allude to the work. line 83 - how does this embedding of a CN.SN in concrete impact the security? Figure 1 does show a valid and viable scenario. It is not motivated how this differs (other than the lack of physical access - line 122) form other deployments e.g., with the devices not entombed within the structure. further mention in 393/399 - again how does this differ from 'normal' deployments. Table 1 - deals with the physical security that is introduced.

The survey and summary of the work is however valid. This could be reframed as considering the security impacts within the embedded case of use. While certainly the physical security aspect is improved, the other problems around power, and accessibility become much more. A further consideration is the increased t power hat may be needed given the non -freespace communications. In essence the core of the work is valid but needs to be presented within an appropriate context.  The conclusion in particular can be more specific around drawing together the pros and cons of embedded systems.

Some other points relating to the document as a whole:

line 50 Consider () - preferred and likely to read easier to most rather than -, or place a space before/after the - (this applies elsewhere in the paper as well)

lines 70-79 - is it worth putting direct reference to section umbers in this rather than the more general approach currently.

line 98 - rather than digital world consider the larger network?

line 147 - again a case for () vs -. there should be no trailing - before the period at end of sentence.

Much of section 3 is background literature and needs to be tied closer to the application and relevance in later sections..

lines 356-359 - rework this. This is quite awkward listing other attacks and then saying they are not considered here. BLE's vulnerability should surely be included in the extended discussion above.

line 430/Table 1 - look at the spacing/hyphenation of reinforced, the split as currently made is unnatural and makes for difficult reading.

Tables 4 and 5 should be carefully considered wrt formatting to remove unnecessary space. (e.g. between the bullet and the item) and in the intercolumn padding). Especially in Table 5 the 3rd column should be shrunk which would allow for expansion of the 4th and thus shortening the table. similarly in the first column the security. text could be rotated as it cuts across multiple cells. It is not ideal to have tables split across pages, especially when headers are not repeated.

Reviewer 2 Report

The paper needs to be enhanced in order to be published in the scientific journal, currently as I see it's a good start, but some work needs to be done.

The English language is good enough.

The paper can be recommended for publication after addressing the following recommendations:

1.1) How the "probability" and "impact" values from Tables 1-3 have been obtained? If this were done by experts assessment, please provide some justification about the confidence of experts in their judgements.

1.2) A needed step of risk management is a risk mitigation strategy. How to mitigate the risks? Any countermeasures? The risk matrices (based on the results of Tables 1-3) with the indication of the current and accepted levels of risks can represent the information better for understanding in comparison to the regular natural language (plain text).

2) Evolving the countermeasure recommendations it could be more interesting and scientifically picturesque to see the extension of Tables 1-3 in the application of IMECA (Intrusion Modes, Effects and Criticality Analysis) applied to the object depicted in Fig.1.

As far as Fig.1 is very general and there is not much information provided in the paper about the internal structure of the object, the application of IMECA (as a technique which can illustrate more information) to a specific object/system with specific hardware will increase the interest and value of the paper.

In case if decided to include IMECA, you can read here more:

- Babeshko, E.; Kharchenko, V.; Gorbenko, A. Applying F(I)MEA-technique for SCADA-Based Industrial Control Systems Dependability Assessment and Ensuring. In Proceedings of the 2008 Third International Conference on Dependability of Computer Systems DepCoS-RELCOMEX, Szklarska Poreba, Poland, 26–28 June 2008; pp. 309–315.

- Androulidakis, I.; Kharchenko, V.; Kovalenko, A. IMECA-Based Technique for Security Assessment of Private Communications: Technology and Training. Inf. Secur. Int. J. 2016, 35, 99–120. [CrossRef]

- Kharchenko, V. Gap-and-IMECA-Based Assessment of I&C Systems Cyber Security. In Complex Systems and Dependability. Advances in Intelligent and Soft Computing, 170; Kharchenko, V., Andrashov, A., Sklyar, V., Siora, A., Kovalenko, A., Eds.; Springer: Berlin/Heidelberg, Germany, 2012; 334p.

- Illiashenko, O.; Kharchenko, V.; Chuikov, Y. Safety analysis of FPGA-based systems using XMECA for V-model of life cycle. Radioelectron. Comput. Syst. 2016, 80, 141–147.

- Babeshko, E.; Kharchenko, V.; Leontiiev, K.; Odarushchenko, O.; Strjuk, O. NPP I&C safety assessment by aggregation of formal techniques. In Proceedings of the 2018 26th International Conference on Nuclear Engineering, London, UK, 22–26 July 2018; pp. 21–26.

3) The title "Security of Wireless Protocols in Critical Environments. Case Study for Communicating Concrete in Civil Engineering" is very general for this paper. Authors are recommended to consider let's say "narrowing" the title to its content.

4) Authors are advised to check the paper against minor mistypes.

Round 2

Reviewer 1 Report

Review Round 2

The authors have in general addressed the concerns raised in the previous round.  The paper is an improved read.

There remain several grammatical/typographical/layout issues to still conder for remediation.

In general, the way tables are structures is awkward and difficult to read at a glance.  The authors should consider some further alteration/optimisation of these.

Keywords: carefully consider if all are needed. Cyber security  for instance should not necessarily be ranked first

lines 49/50 - the use of the term twins here does not necessarily make sense. given its lack of prior introduction. Possibly replace "digital models/twins " with virtual equivalents or virtual representations

line 78 - consider removal of the () as the sentence reads fine without them and removes visual clutter.

line 80 - should there not be a citation to the McBIM project here?

line 108 - You are explicitly stating int he title that the paper deals with the security aspects of communications, this line does not fit. If this refers to the data, then this needs to be made more explicit.

line 115 - Bluetooth Low Energy is abbreviated as BLE earlier in line 78. Is it worth using the acronym here, as it stands it is not used in the rest of the text?

line 144 consider replacing buried with encapsulated. as they are not necessarily below ground and can be in structures such as pillars.

line 186 reinforced concrete structures?

line 430 - Citation for FMECA? Given the readership of the journal is not necessarily all structural engineers.

Table 1 - consider using centred text in columns 2 & 3 to match the first, current ragged structure is not optimal.

line 523 - proprietary driver software?

Table 4 consider reformatting to reduce size, improve readability.

Table 5 - Fix formatting inconsistency with Numeric comparison

References appear to be correctly formatted and relevant to the research.

Reviewer 2 Report

The recommendations were properly addressed, so the paper can be recommended for publication in the present form.
